# Interstitial Lung Disease in Children: “Specific Conditions of Undefined Etiology” Becoming Clearer

**DOI:** 10.3390/children9111744

**Published:** 2022-11-14

**Authors:** Santiago Presti, Giuseppe Fabio Parisi, Maria Papale, Eloisa Gitto, Sara Manti, Salvatore Leonardi

**Affiliations:** 1Pediatric Respiratory Unit, Department of Clinical and Experimental Medicine, University of Catania, Via Santa Sofia 78, 95123 Catania, Italy; 2Neonatal Intensive Care Unit, Department of Human Pathology of Adult and Childhood Gaetano Barresi, University of Messina, Via Consolare Valeria, 1, 95122 Messina, Italy; 3Pediatric Unit, Department of Human Pathology of Adult and Childhood Gaetano Barresi, University of Messina, Via Consolare Valeria, 1, 95122 Messina, Italy

**Keywords:** neuroendocrine cell hyperplasia of infancy (NEHI), pulmonary interstitial glycogenosis (PIG), children’s interstitial lung disease

## Abstract

Background: Children’s interstitial lung disease (chILD) is a rare group of pediatric lung diseases affecting the lung interstitium diffusely. In this work, we focused our attention on a specific infant group of chILD, also known as “specific conditions of undefined aetiology”, including pulmonary interstitial glycogenosis (PIG) and neuroendocrine cell hyperplasia of infancy (NEHI). Methods: PubMed was searched to conduct this narrative review. We searched for articles in English using the following keywords: (1) neuroendocrine cell hyperplasia of infancy; (2) NEHI; (3) pulmonary interstitial glycogenosis; (4) PIG; (5) chILD. Results: An increasing interest and insight into these two conditions have been reported. The updated literature suggests that it is possible to look at these disorders as a continuum of diseases, rather than two different entities, since they share a pulmonary dysmaturity. Conclusions: NEHI and PIG are featured by dysmaturity of airway development and consequent respiratory distress. Understanding the underlying pathogenic mechanisms would lead to identifying new targeted therapies to ameliorate the mortality and morbidity of these rare conditions.

## 1. Introduction

Children’s interstitial lung disease (chILD) is a term referring to a rare group of pediatric lung diseases affecting the lung interstitium, predominantly. Recently, the term “interstitial” has been changed to “diffuse”, since it is more appropriate to describe these conditions, as they can also involve distal small airways and/or terminal bronchioles, and alveoli [1,2,3]. Several classifications have been proposed based on the age at onset [1] or the aetiology [2]. The central role of ILD’s pathogenesis is the aberrant activation of alveolar epithelium and mesenchymal cells. Several mechanisms are involved, such as injuries of alveolar epithelial cells (AEC) and incorrect response to injury of alveoli, leading to aberrant lung repair and progressive fibrosis [2]. A multiple-hit model has also been proposed, including both genetic and non-genetic factors [4]. The pathological mechanisms lead to changes in cellular phenotype and function. The epithelial cells would acquire mesenchymal cell phenotypic and functional characteristics; therefore, the production of collagen and extracellular matrix components would lead to an aberrant distortion of the interstitium with consequent alteration of gas exchange [2,4,5,6,7,8].

In this work, we reviewed the most up-to-date findings of a specific group of chILD, named “specific conditions of undefined aetiology”, including pulmonary interstitial glycogenosis (PIG) and neuroendocrine cell hyperplasia of infancy (NEHI) [1].

## 2. Materials and Methods

PubMed was searched to conduct this narrative review. We searched for articles in the English language, and no time limit was adopted. The following keywords were used: (1) neuroendocrine cell hyperplasia of infancy; (2) NEHI; (3) pulmonary interstitial glycogenosis; (4) PIG; (5) children’s interstitial lung disease. Two independent reviewers performed data extraction using standard templates. Articles were excluded by title, abstract, or full text for irrelevance to the investigated issue.

## 3. Results

The most updated literature suggests that PIG and NEHI are characterized by pulmonary dysmaturity; thus, it could be more appropriate to look at these two diseases as an expression of pulmonary dysmaturation. Accordingly, the term “spectrum of pulmonary dysmaturation disorders” has been proposed to refer to these clinical entities [7,8].

### 3.1. Neuroendocrine Cell Hyperplasia of Infancy (NEHI)

In 2005, Deterding et al. described in the lung biopsies of a group of surviving children over five years of age an increased number of neuroendocrine cells (NECs); thus, authors included NEHI within the group of chILDs [9] Table 1.

During intrauterine life, NECs are expressed in the distal airways promoting branching morphogenesis, epithelial and mesenchymal cell proliferation, and surfactant secretion. They induce airway epithelial and mesenchymal cells proliferation, and alveolar type II cells differentiation. Conversely, during the neonatal period, NECs decline rapidly [10,11,12]. It is still unclear if the presence and the activity of NECs represent a primary pathological mechanism or if they reflect a secondary reaction to other conditions [11].

In 2010 Popler et al. highlighted the presence of NEHI in siblings of four families, suggesting the involvement of genetic factors in the pathogenesis of chILD [13].

Young et al. found in a patient with NEHI, and in a family with a history of childhood lung disease, a heterozygous substitution in NKX2.1 gene that encodes TTF-1 [11].

New insights suggest that NEHI consists of dysmaturity of the foetal airway, as the detection of bombesin-positive cells, commonly expressed during physiological lung organogenesis, is usually considered a marker of dysmaturity [7,14].

The extension of neuroendocrine cell prevalence in respiratory bronchioles leads to air-trapping and small airway obstruction [10]. It has been hypothesized that the symptoms experienced by the patients could result from the production of bronchiolar constrictors, such as serotonin, bombesin and calcitonin, by the overactive pulmonary neuroendocrine cells [10,15]. Affected infants present significantly low tidal volumes with high minute ventilation and airflow limitation, showing low forced vital capacity (FVC) due to air-trapping and reduction in the forced expiratory volume (FEV1). Functional residual capacity, residual volume and residual volume/total lung capacity are usually above the normal range. Generally, post-bronchodilator measurements do not show improvements [16,17,18]. The severity of small airway obstructions is related to the NEC’s prominence [10].

The disease onset is usually within the first year of life. Infants present stable general conditions with chronic tachypnea and intercostal retractions but with severe worsening of clinical stability after viral infections [16]. The most frequent pulmonary manifestations are tachypnea, hypoxemia (>90%), retractions and crackles (>80%). Not common are clubbing, coughing or wheezing at the onset. Failure to thrive and developmental delays are usual non-pulmonary signs [19,20]. A recent review showed that the median age of symptomatic onset was 3 months, while the diagnosis commonly occurred at the age of 6 months [21].

Multiple operative flow-chart and diagnostic approaches are proposed for the diagnosis of chILDs [2,22,23]. Upon clinical suspicion of a chILD, the starting point is performing a high-resolution computer tomography (HRCT) as soon as possible. It can be suggestive for a specific diagnosis or not. Noninvasive approaches such as genetic tests are suggested if the patient is stable and the imaging is not suggestive. However, if the patient is not stable, invasive tests such as lung biopsy must be considered. In the case of NEHI, diagnosis is performed by clinical and radiological findings and, commonly, the biopsy is not necessary [19,24]. Liptzing et al. performed a NEHI clinical score, a sensitive tool for clinically evaluating NEHI [19]. Standard X-ray usually does not help, since it highlights the signs of interstitial lung disease without good specificity. HRCT findings, instead, reveal ground-glass opacities predominantly in the middle lobe and lingula and mild air-trapping with mosaic attenuation [24,25,26]. HRCT has 78% specificity and 100% sensitivity [25]. Ground-glass opacities could be considered as a biomarker of NEHI severity [27]. Other HRCT pathological findings such as consolidation, bronchial wall thickening, bronchiectasis, linear and reticular opacity, nodules, and honeycombing are rarely described [25]. No other radiological techniques are useful in the diagnosis of NEHI.

Lung biopsies typically show an increase in alveolar macrophages, smooth muscle hyperplasia of bronchioles, and the presence of NECs within distal airways, marked by immunostains against bombesin and serotonin [9,28]. Usually, alveoli show typical structures without any significant fibrosis. NECs are detectable almost in 70% of bronchioles and at least one individual airway, with ≥10% of NECs without active proliferation [10,15]. The % NEC area is twofold more remarkable in the proximal bronchioles [10]. NECs release bombesin, probably involved in the pathogenesis of small airway obstruction [29]. Mastej et al. demonstrated that lung and airway appearances in patients with NEHI presented an increased anteroposterior diameter, suggesting that also this criterion might be considered in the diagnostic criteria [30]. As demonstrated by Doan et al., the level of serum glycoprotein KL-6 might be a useful biomarker to distinguish NEHI from more severe neonatal surfactant metabolism disorders. In fact, NEHI patients present normal KL-6 levels compared to patients with SP-C and ABCA3 [31].

Since few cases are reported in the literature, evidence about the therapy is sparse. Studies suggest that there is no benefit using either systemic or inhaled corticosteroids; thus, therapy is normally represented only by supportive oxygen treatment with very good prognosis, often with a complete recover [15,32]. The lung injury may persist even for years with a slow resolution, and patients may constantly be air-trapped. Usually, clinical symptoms are progressively resolved, presenting respiratory exacerbations characterized by increased air-trapping, often triggered by viral infections [33]. Lukkarinen et al. reported the association of non-atopic asthma in children with a history of NEHI, underlying the importance of thinking about chILD and specifically NEHI in children with chronic respiratory symptoms [15]. Adult follow-up is rarely reported since NEHI is a relatively new entity. Although clinical improvement occurs, mild radiologic abnormalities might persist over time [34]. NEHI patients might experience sleep disorders such as obstructive and central sleep apnea, hypoxemia, decreased sleep efficiency and increased periodic limb movement disorder [35].

**Table 1 children-09-01744-t001:** NEHI main features.

Age at onset	Within the first year	[9,21]	HRCT findings	Specific: ground-glass opacities predominantly in the middle lobe and lingula and mild air-trapping with mosaic attenuation	[24,25,26,27]
Etiology	Unknown	[11]	Diagnosis	Clinical and radiological	[2,22,23]
Anomalies	Neuroendocrine cells in respiratory bronchioles	[12]	Biopsy	Mild increase of alveolar macrophages and smooth muscle hyperplasia of bronchioles and the presence of NECs within distal airways, marked by immunostains against bombesin and serotonin	[9,28]
Lung dynamic alterations	Low tidal volumes, high minute ventilation, low forced vital capacity (FVC). Functional residual capacity, residual volume, and residual volume/total lung capacity are above the norm. Generally, post-bronchodilator measurements do not show improvements	[10,15]	Therapy	Supportive oxygen treatment	[15,32]
Signs and symptoms	Chronic tachypnea, hypoxemia (>90%), retractions and crackles (>80%), failure to thrive and developmental delays	[16,17,18,19,20]	Prognosis	Very good, often with a complete recover	[15,33,35]

### 3.2. Pulmonary Interstitial Glycogenosis (PIG)

Pulmonary interstitial glycogenosis (PIG) is a form of interstitial lung disease of unknown origin, and occurring in infants. In 2002, Canakis et al. first described a series of seven infants in the first month of life with tachypnea, hypoxemia and diffuse interstitial infiltrates. The lung biopsies showed an expanded interstitium by spindle-shaped cells containing periodic acid–Schiff positive–diastase, a labile material consistent with glycogen. Therefore, the term “pulmonary interstitial glycogenosis” has been proposed [36] Table 2.

No abnormal glycogen deposition is found in other districts; thus, this clinical entity is not considered a glycogen storage disease [8].

The aetiology is still unknown. It remains unclear if a developmental anomaly with aberrant differentiation causes the presence of cytoplasmic glycogen in epithelial cells or if it is sustained by a reactive process secondary to underlying conditions. Accordingly, cytoplasmic glycogen in epithelial cells early in fetal lung development is commonly reported [36,37,38]. Aggregates of glycogen are present during lung development, but they disappear in post-partum pulmonary interstitial cells [7]. These insights lead to considerations of PIG as a result of pulmonary dysmaturity, in which lung parenchyma maintains fetal characteristics [7].

Patients typically present neonatal distress and hypoxemia without signs of infection [37], and often require mechanical ventilation. Rarely, infants can present tardive symptoms such as pulmonary hypertension without cardiac anomalies [39]. In a recent study by Seidl et al., a significant percentage of PIG patients presented comorbidities: 72.7% of patients presented congenital heart defects, 18.1% presented metabolic diseases, and 9% of patients presented heterotaxy syndrome due to genetically confirmed primary ciliary dyskinesia [37]. The association between PIG and extrapulmonary anomalies might support the theory of a development abnormality [40]. Other studies highlighted the association with comorbidities such as airway malacia, alveolar simplification, congenital diaphragmatic hernia, connective tissue disorder, seizures, aspiration, autism, single kidney, septo-optic dysplasia, and diaphragmatic eventration [41,42]. As mentioned for NEHI, symptoms are not specific; typically, PIG patients present tachypnea and hypoxemia. Two thirds of patients require neonatal resuscitation, including non-invasive ventilation and/or invasive mechanical ventilation [40,42].

Differently to NEHI, HRCT does not show pathognomonic signs. The most common signs are ground-glass opacities (absolutely the most common), cystic lucencies, predominantly in posterior lung fields, consolidations, interlobular septal thickening, linear opacities, mosaic attenuation and architectural distortion. Less commonly, it is possible to identify hyperinflated secondary lobe, emphysema, atelectasis and crazy paving patterning. Since there is heterogeneity of characteristics, there is a significant overlap with other interstitial lung diseases, such as surfactant dysfunction mutations. For this reason, histopathologic findings are essential for diagnosis [37,41,43].

Lung biopsy shows glycogen accumulation and abnormal proliferation of mesenchymal cells [1,39]. The histopathological pattern is represented by patchy (less than half of the interstitial tissue) or diffuse (more than half of the interstitial tissue) distribution [40]. The diffuse expression of vimentin and focal smooth muscle actin positivity might be a sign of a fibroblast phenotype. In addition to glycogen, droplets of neutral lipid are present, configuring a phenotype of lipofibroblasts [44]. By ultrastructure, PIG cells present sparse organelles without specific features that indicate differentiation, considering these cells primitive. Since the expression of CD44, CD73, CD90 and CD105 and the lack of expression of hematopoietic markers, it has been hypothesized that PIG cells are lung-resident mesenchymal stem cells. Thus, these cells would present the ability to differentiate into multiple cell types, such as adipocytes, osteocytes and chondrocytes [45].

In addition to oxygen supplementation, patients are treated with systemic corticosteroids. The response to steroid therapy is still controversial, and more studies are needed [42]. Even if no active inflammation is normally evinced in lung biopsies, the rationale of corticosteroids might be justified by the aim to promote tissue maturation, acceleration of lipofibroblasts apoptosis, alveolar remodelling and surfactant production during the neonatal period [44,46].

**Table 2 children-09-01744-t002:** PIG main features.

Age at onset	Neonatal or first months	[7,8,36]	HRCT findings	Not specific: ground-glass opacities, cystic lucencies, both predominantly in posterior lung fields, consolidations, interlobular septal thickening, linear opacities, mosaic attenuation and architectural distortion	[37]
Etiology	Unknown	[7,36,37,38]	Diagnosis	Biopsy needed	[37]
Anomalies	Spindle-shaped cells containing periodic acid–Schiff positive–diastase labile material consistent with glycogen and abnormal proliferation of mesenchymal cells that expand the interstitium	[36,38,46]	Biopsy	Interstitium expanded by spindle-shaped cells containing periodic acid–Schiff positive–diastase labile material consistent with glycogen and abnormal proliferation of mesenchymal cells. Histopathological pattern represented by patchy or diffuse distribution. Diffuse expression of vimentin and focal smooth muscle actin positivity. In addition to glycogen, droplets of neutral lipid are present.	[8,44,46]
Lung dynamic alterations	Mostly obstructive pattern while restrictive pattern is possible but less frequent		Therapy	Oxygen supplementation and systemic corticosteroids	[42]
Signs and symptoms	Tachypnea and hypoxemia. Two thirds of patients require neonatal resuscitation, including non-invasive ventilation and/or invasive mechanical ventilation	[37,40,42]	Prognosis	Variable, half of patients become asymptomatic after 2 or 3 years from the diagnosis	[47,48]

Reports show that the lesional mesenchymal cells may have transient proliferative capacity; accordingly, the prognosis is good but further studies are needed in order to describe the long-term history of this disease [47]. Studies suggest that about half of patients become asymptomatic after 2 or 3 years from the diagnosis, and another half still present some respiratory symptoms such as tachypnoea and reduced exercise tolerance. Only about 10% of patients died of acute respiratory insufficiency due to viral infections. There is no correlation between the severity of clinical, radiological or histopathologic characteristics and outcomes [37]. In a long-term follow-up (median 12 years), Sardón et al. highlighted that all patients were asymptomatic, but respiratory function tests showed abnormalities in almost all patients. Most of subjects presented an obstructive pattern, while a restrictive pattern was less represented. Additionally, HRCT performed after an average of 6.5 years revealed a not-complete improvement of the ground-glass, with the persistence of relevant alterations [48].

## 4. Discussion

Even if the aetiology is still unknown, it seems reasonable that NEHI and PIG originate during fetal lung development due to pulmonary dysmaturity [7,36,37,38,49]. In fact, glycogen aggregates are present during the development of the lung, and it is possible to find NECs in the distal airways promoting branching morphogenesis, epithelial and mesenchymal cell proliferation, and surfactant secretion [7,10]. Although the origin seems similar, the two entities differ in clinical and radiological characteristics. The onset of NEHI is generally within the first year of life, and, typically, infants present stable clinical conditions with chronic tachypnea and intercostal retractions, but with severe worsening of clinical stability after viral infections [16]. However, PIG patients typically present neonatal or early distress and hypoxemia without signs of infection [37], often requiring mechanical ventilation. Rarely, infants present tardive symptoms such as pulmonary hypertension without cardiac anomalies [39,50]. The two entities differ by imaging characteristic. In NEHI, HRCT typically reveals ground-glass opacities predominantly in the middle lobe and lingula, and mild air-trapping with mosaic attenuation [19,24,25,26]. Conversely, since HRCT is not specific in PIG, a lung biopsy must be performed. Usually, it reveals the accumulation of glycogen and abnormal proliferation of mesenchymal cells [1,37,39,51] Figure 1. NRHI and PIG also differ by treatment: while NEHI patients usually present a progressive spontaneous resolution of clinical symptoms, in PIG, patients are treated with systemic corticosteroids even if the response is still under debate.

## 5. Conclusions

chILD represents rare respiratory disorders, including highly heterogeneous etiologies, severities and prognosis. An early diagnosis is mandatory, and it is essential to know the correct diagnostic approach. New insights into NEHI and PIG highlight the central role of dysmaturity in the pathophysiology of these rare conditions. Both entities share the same possible origin: dysmaturity of the airway development with consequent respiratory distress. Understanding the underlying pathogenic mechanisms would lead to identifying new targeted therapies to ameliorate the mortality and morbidity of these rare conditions.

## Figures and Tables

**Figure 1 children-09-01744-f001:**
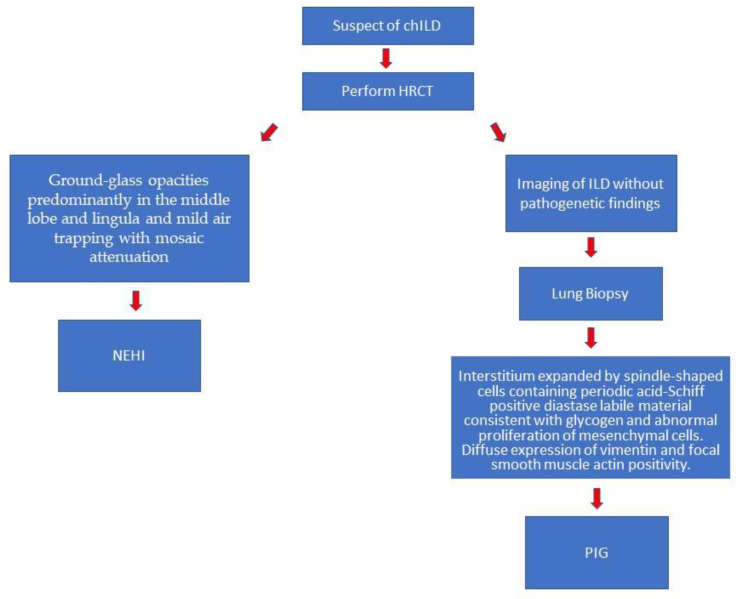
Diagnostic flow-chart.

## Data Availability

Not applicable.

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
