# Peer review of "Interstitial Lung Disease in Children: “Specific Conditions of Undefined Etiology” Becoming Clearer"

_children, 2022, doi:10.3390/children9111744_

Round 1
Reviewer 1 Report
This manuscript reviewed the most up-to-date findings of a specific group of chILD more prevalent in infancy named “specific conditions of undefined etiology”. It has good guiding value for clinical work. However, the introduction, materials and methods, and results are generally in the form of article, and the review is unnecessary. And the results of the article are the results of literature, not original. In addition, it is recommended to add quotation marks to the references cited in the tables.
Author Response
Reponse: Many thanks for your kind words. However, we have to adapt the paper in accordance with the Journal guidelines. As suggested, we added quotation marks to the references cited in the tables.
Reviewer 2 Report
Presti and colleagues provide a concise summary of PIG and NEHI. This review requires significant improvements on English language before being published by Children.
Author Response
Reponse: As suggested, the paper has been reviewed with the help of native English language.
Reviewer 3 Report
Presti and colleagues submitted a narrative review on interstitial lung disease in childhood. They summarized well each lung disease entity and provide two important tables. I only have some minor feedbacks: Please adapt the abstract section to the journal requirements. Please provide a graphical abstract or a figure by summarizing e.g., a diagnostic work-up strategy for physicians in case of clinical suspicion. Please add a discussion section.
Author Response
Reponse: Many thanks for your kind words. However, we have to adapt the paper in accordance with the Journal guidelines. Also, we added a flow-chart summarizing the diagnostic work-up strategy for physicians. Discussion section has been also added.
Round 2
Reviewer 2 Report
The manuscript still has several grammatic errors and could be improved
science part is sound
Author Response
Following your suggestions, the manuscript has been improved
